# Inflammatory Blood Signature Related to Common Psychological Comorbidity in Chronic Pain

**DOI:** 10.3390/biomedicines11030713

**Published:** 2023-02-27

**Authors:** Bianka Karshikoff, Karin Wåhlén, Jenny Åström, Mats Lekander, Linda Holmström, Rikard K. Wicksell

**Affiliations:** 1Department of Social Studies, University of Stavanger, 4036 Stavanger, Norway; 2Department of Clinical Neuroscience, Karolinska Institutet, 17177 Stockholm, Sweden; 3Pain and Rehabilitation Centre, and Department of Health, Medicine and Caring Sciences, Linköping University, 58183 Linköping, Sweden; 4Medical Unit Medical Psychology, Theme Women’s Health and Allied Health Professionals, Karolinska University Hospital, 17176 Solna, Sweden; 5Stress Research Institute, Department of Psychology, Stockholm University, 106 91 Stockholm, Sweden; 6Pain Clinic, Capio St. Göran Hospital, 11219 Stockholm, Sweden

**Keywords:** biomarkers, cytokines, inflammation, chronic pain, psychological comorbidity, pain intensity, depression, anxiety

## Abstract

Chronic pain is characterized by high psychological comorbidity, and diagnoses are symptom-based due to a lack of clear pathophysiological factors and valid biomarkers. We investigate if inflammatory blood biomarker signatures are associated with pain intensity and psychological comorbidity in a mixed chronic pain population. Eighty-one patients (72% women) with chronic pain (>6 months) were included. Patient reported outcomes were collected, and blood was analyzed with the Proseek Multiplex Olink Inflammation Panel (Bioscience Uppsala, Uppsala, Sweden), resulting in 77 inflammatory markers included for multivariate data analysis. Three subgroups of chronic pain patients were identified using an unsupervised principal component analysis. No difference between the subgroups was seen in pain intensity, but differences were seen in mental health and inflammatory profiles. Ten inflammatory proteins were significantly associated with anxiety and depression (using the Generalized Anxiety Disorder 7-item scale (GAD-7) and the Patient Health Questionnaire (PHQ-9): STAMBP, SIRT2, AXIN1, CASP-8, ADA, IL-7, CD40, CXCL1, CXCL5, and CD244. No markers were related to pain intensity. Fifteen proteins could differentiate between patients with moderate/high (GAD-7/PHQ-9 > 10) or mild/no (GAD-7/PHQ-9 < 10) psychological comorbidity. This study further contributes to the increasing knowledge of the importance of inflammation in chronic pain conditions and indicates that specific inflammatory proteins may be related to psychological comorbidity.

## 1. Introduction

Chronic pain is a challenging disorder to study given the complexity of the underlying mechanisms, which include the immune system, and the high psychological comorbidity [1,2]. Many neuroimmune and psychological mechanisms [3,4] have been identified that seem to play a role in the development and maintenance of pain that persists without a corresponding peripheral sensory input. Chronic pain diagnoses are rarely distinct entities. The diagnostic criteria were recently revised by the International Association for the Study of Pain (IASP) to reflect modern findings and facilitate an adequate distinction between functionally different types of chronic pain. In the ICD-11, pain syndromes are divided into mechanistic groups that encompass nociceptive pain (actual tissue damage and nociceptor activation), neuropathic pain (lesions to the somatosensory nervous system), and nociplastic pain (altered pain perception despite no discernable damage) [5,6]. Many pain syndromes do, however, include a mixture of these processes. Importantly, individuals with chronic pain commonly present with psychological comorbidities, including anxiety, depression, fatigue, disturbed sleep, and cognitive deficits [3,4]. These comorbidities can often affect the person’s quality of life and functioning as much as the pain intensity itself [7] and may negatively influence the effects of treatment. In turn, pain is one of the most common comorbid problems in illness in general and is one of the most significant problems accounting for years lived with disability, according to WHO [8]. Chronic pain encompasses many fundamental problems that characterize non-communicable, complex disorders. It is profoundly costly for society and often devastating for the individual [5]. It affects a large part of the population, but increasing age and female sex increase the risk [7]. Several treatment options may be helpful for many individuals suffering from pain, but often, no complete cure can be offered [5].

The understanding of chronic pain has increased in the last few decades, and modern pain treatment incorporates psychological aspects of the disorder [5], acknowledging that pain is not solely a dysfunction of the nervous system but a multisystem problem. Lately, the focus has been on the immune system, highlighting neuroimmune interactions and inflammatory mechanisms in chronic pain conditions [1,2,3,4,9,10,11,12,13]. This line of research suggests that the immune system is involved in both central and peripheral sensitization of pain neurons [2], in the development and priming of the pain system [1], in sex differences in prevalence [12,13], and in the bidirectional interplay between pain and psychological wellbeing [3,4]. Systemic low-grade inflammation has been indicated in several pain diagnoses [14,15,16,17,18], similar to the findings for inflammation and depression [19,20,21,22]. Pain diagnoses are generally symptom-based due to a lack of clear pathophysiological factors and reliable and relevant biomarkers. Challenges in identifying target molecules for pharmacological treatments of chronic pain have led to system-based enquiries, such as proteomics studies or larger targeted multiplex analyses, to study networks of expressed proteins in different chronic pain conditions. Gomez-Varela et al. suggest the term protein disease signature (PDS) to describe networks or groups of proteins related to chronic pain in general, to specific chronic pain syndromes, or phases or aspects of persistent pain [23]. For now, no PDS specific enough to serve as a biomarker for diagnosis or progression of chronic pain has been identified. In PDS studies, many markers are studied in an exploratory and data-driven fashion. Such studies may reveal the biological underpinnings of this common and severe disorder that are unknown to date and can subsequently be investigated in directed, experimental studies. Most pain studies using this approach have been cross-sectional, comparing defined pain populations with healthy controls, assuming that the differences between groups reflect a specific pattern in the chosen biological fluid related to that particular pain disorder [24,25,26,27,28,29,30,31,32,33]. The focus is usually on pain intensity, as pain research ultimately aims to eradicate the pain [34,35,36,37,38]. Some have pointed to correlations between PDS and pain severity, but most have not. The few studies investigating psychological comorbidity in pain with a proteomic approach indicate that the related PDS differ for pain sensitivity (pressure pain thresholds), clinical pain (pain intensity), or psychological distress (anxiety and depression using total HADS) in pain disorders of widespread pain [32,38,39]. Wåhlén et al. show that distinct protein networks largely do not overlap and are related to pain or psychological distress, respectively, in fibromyalgia and chronic widespread pain. This exemplifies the complexity of pain syndromes regarding the relationship between biological networks and clinical outcomes.

In this study, we make assumptions that influence analyses and interpretations. The first is that chronic pain is multifaceted. Pain intensity is the primary treatment outcome, but the disorder encompasses levels of suffering beyond pain intensity, such as depression and anxiety. The second is that some mechanisms will be specific for certain pain disorders (e.g., mechanisms accompanying the damage of nerves in neuropathic pain), but some are potentially overarching, such as immune dysfunction, and involved in several pain disorders. We thus focus on proteins belonging to the inflammatory system from an easily accessible tissue (blood). We also proceed from the broad knowledge assimilated in the past decade on inflammation-driven mood changes in experimental [40,41,42,43] and clinical settings and in population studies [19,44,45,46,47]. These studies suggest that an activated immune system may drive depressive mood and anxiety, as well as heightened pain sensitivity. While inflammatory effects have been intensely investigated for depression [19,21,22], it should be noted that higher levels of common inflammatory markers (e.g., CRP and IL-6) in clinical anxiety diagnoses are inconsistent, most consistently represented in post-traumatic stress disorder [48,49,50]. For chronic pain, PDS research has increased in recent years (e.g., [32,51]). Thus, this study attempts to explore the relationship of 77 inflammatory markers from a well-established and well-used protein marker panel and explore potential subgroups of a chronic pain patient sample. This study is an exploratory assessment within a project designed to investigate the effects of ongoing inflammation in pain, pain comorbidity, and behavioral treatment (https://openarchive.ki.se/xmlui/handle/10616/48067 (accessed on 8 January 2023)) [52]). The sample consists of mixed pain for pragmatic reasons and to optimize ecological validity, i.e., a clinically representative pain population. We asked which inflammatory markers were related to the pain itself versus common psychological comorbid problems in a clinically representative pain population and if some markers can distinguish patient subgroups based on their pain and/or psychological comorbidity. This investigation aims to add to the growing literature on the involvement of inflammation in chronic pain and problematize the different subcomponents of this complex disorder.

## 2. Materials and Methods

### 2.1. Study Participants and Procedure

Participants were recruited between 2016 and 2018 as patients at the Behavioral Pain Medicine Treatment Services at the Karolinska University Hospital (Sweden) or via advertisements (self-referral). The inclusion criteria for the study were men and women ≥ 18 years of age; pain duration > 6 consecutive months; unresponsiveness to general treatments; stable medication the last two months; and must understand/read fluent Swedish. Exclusion criteria were pregnancy, breastfeeding or having given birth within the previous year, and hemophilia. Further, patients participating in a concurrent cognitive behavioral therapy (CBT)-based treatment, such as acceptance and commitment therapy (ACT), as well as patients who presented with severe psychiatric comorbidity that required immediate assessment or treatment (e.g., high risk of suicide, psychotic symptoms), were excluded. Finally, if a spontaneous improvement in their pain could be expected, participants were also excluded (see [52] using the same cohort). The initial screening processes resulted in 113 participants. After evaluation of eligibility according to the inclusion and exclusion criteria, 81 subjects (women *n* = 58 (72%), men *n* = 23 (28%)) with chronic pain (>6 months) were included in the study. For an overview of the recruitment process and reason for exclusion, see the flow chart in Figure 1.

### 2.2. Patient-Reported Outcomes (PROMs)

Self-reported Swedish questionnaires, including several validated measures, were used to record patient-reported outcomes and were filled out digitally by each participant in connection to blood sampling.

#### 2.2.1. Anthropometric Parameters

Sex, age (years), weight (kg), and height (cm) were recorded via self-assessment, and body mass index (BMI) was calculated (kg/m^2^).

#### 2.2.2. Pain Parameters

Each subject self-reported their present pain intensity (whole body) with the numeric rating scale (NRS). The NRS scale ranges from 0 to 10, where 0 equals no pain and 10 equals the worst imaginable pain. To characterize what type of pain was present, each subject reported present pain (yes or no) in the following area: head, facial, teeth/jaw, neck, back, chest, abdomen, genital, hands, legs, feet, and entire body. These scores were used to determine whether the pain was of local or of a more widespread origin. A score of reported pain in ≥3 areas was regarded as widespread pain. Local pain was classified if the pain was present in only one area or two areas in the same body part, e.g., leg and feet. Mean painful areas were calculated and reported as the variable *pain area number*.

Pain duration (years) for each area was reported and calculated as mean pain duration over all sites.

The Pain Disability Index (PDI) consists of 7 items and assesses how much impact the perceived pain has regarding hindering everyday life activities [53]. Each item is summarized into a total score ranging from 0 to 70, where a higher score indicates greater disability due to pain.

The Pain Interference Index (PII) measures how pain influences functioning. It consists of 6 items, where the total score is summarized, ranging from 0–36 [54]. A higher score indicates more pain interference.

The Pain Catastrophizing Scale (PCS) measures catastrophizing thoughts and consists of 13 items divided into three subscales that measure rumination, helplessness, and magnification. Each subscale is summarized to a total score ranging from 0 to 52, where a higher value indicates worse catastrophizing thoughts [55].

#### 2.2.3. Psychological Parameters

The Patient Health Questionnaire-9 (PHQ-9) was used to assess present depressive symptoms. The questionnaire consists of nine items, with each item scoring 0–3 points. The total sum score (maximum 27) was calculated, and cutoff values for depressive symptoms were set to: 0–4 = none, 5–9 = mild, 10–15 = moderate, and 15–27 = moderate/severe [56].

The Generalized Anxiety Disorder 7-item scale (GAD-7) was used to assess symptoms of anxiety. The GAD-7 consists of 7 items, with each item scoring 0–3 points. The total sum score (maximum 21) of all seven items was calculated, and cutoff values for anxiety symptoms were set to: 0–4 = none, 5–9 = mild, 10–14 = moderate, and 15–21 = severe [57].

#### 2.2.4. Other Clinical Parameters

Self-rated health (SRH-5) was used to evaluate how the participants rated their general health status. The questionnaire consists of a single item and is scored from 0 to 5. For facilitating the interpretation, the score was reversed when summarized so that 0 = poor health and 5 = good health [58].

The Insomnia Severity Index (ISI) was used to assess insomnia problems. It consists of seven items, with a total score ranging from 0 to 28 [59]. A higher score indicates greater insomnia problems.

The Perceived Stress Scale (PSS) was used to evaluate general perceived stress levels. It consists of 10 items, with each item scoring 0–4. The total score from all items is summarized to generate a score ranging from 0 to 40 [60]. The higher the score, the larger the perceived stress level.

Short Form-12 Health Survey (SF-12) is a shorter form of the SF-36 questionnaire measuring health-related quality of life summarized in two dimensions; a physical component score (PCS-12) and a mental component score (MCS-12) [61]. PCS-12 and MCS-12 consist of several questions with different scoring, which generate a total sum score ranging from 0 to 100. The higher the score, the better the patient’s physical and mental wellbeing.

### 2.3. Blood Sampling

Non-fasting venous blood sampling using EDTA tubes was performed between 8 and 12 a.m., and samples were processed within two hours. No more than 50 mL of blood was drawn. Routine analysis measures of C-reactive protein (hs-CRP), erythrocyte sedimentation rate (sr-ESR), and cortisol in blood were performed at Karolinska University Hospital. If elevated CRP values (>10 mg/L) were detected, the patients were informed of the results by a physician. Plasma samples were further aliquoted and stored in Karolinska University Hospital Biobank at −80 °C and went through one thawing cycle before analysis. For further details, see [52].

### 2.4. Inflammatory Cytokine Panel Analysis

Plasma samples from each participant were analyzed on the Proseek Multiplex Olink Inflammation Panel using the Proximity Extension Assay technology provided by OLINK (Bioscience Uppsala, Sweden) according to the manufacturer’s instructions (for information on all cytokines included in the panel, see www.olink.com/products/inflammation/ (accessed on 8 January 2023)) [62,63]. The inflammatory panel included 92 cytokines, chemokines or growth factors. Eighteen proteins were excluded due to missing values in >60% of the samples (IL-17C, IL-20RA, IL-2RB, IL-1 alpha, IL-2, FGF-5, IL-22RA1, Beta-NGF, IL-24, IL-13, ARTN, IL-20, IL-33, IL-4, LIF, NRTN, IL-5, and TSLP), resulting in 77 proteins from the OLINK panel included in the downstream statistical analysis. The generated data are expressed as normalized protein expression (NPX), and values of NPX are acquired by normalizing Cq values against extension control, interplate control and a correction factor. The dataset can be used for statistical multivariate analysis and expresses relative quantification between samples but is not an absolute quantification.

### 2.5. Statistical Analysis

Univariate and descriptive statistics were performed in GraphPad Prism version 7.0.5 for Windows (GraphPad Software, San Diego, CA, USA). For comparison between subgroups (PROMs and specific protein markers), the Kruskal–Wallis test followed by post hoc Dunn’s multiple comparison tests were applied. No further correction for multiple testing was performed for univariate statistics. All analyses used a *p*-value ≤ 0.05 as the significance level. Multivariate data analysis (MVDA) was performed in SIMCA version 16 (Sartorius Stedim Biotech, Umeå, Sweden). The procedure for computing MVDA has been described in detail elsewhere [24,30,32,33,64]. MVDA is often used when large omics panels are investigated and multivariate correlation between variables exists. Since MVDA accounts for the internal correlation structures of a dataset (analyzing all variables simultaneously), no further correction for multiple testing is needed, as is often necessary using other type of tests such as mixed models or univariate comparison. Before starting the MVDA, NPX data were log2 transformed and scaled with unit variance, following principal component analysis (PCA) in combination with hierarchical clustering analysis (HCA), orthogonal partial least-square analysis (OPLS), and based on the subgroups identified in the HCA, OPLS–discriminant analyses (OPLS-DA).

#### 2.5.1. Principal Component Analysis (PCA)

PCA is an unsupervised dimensional reduction method used to investigate correlation structures among several variables (e.g., inflammatory proteins) and a set of observations (each individual subject). In this study, we have used inflammatory proteins as x-variables (*n* = 77) and included 81 observations (subjects) consisting of chronic pain patients. Initially, the PCA was used to detect potential outliers in the dataset using Distance to model X (DModX) and Hotelling’s T2 in SIMCA [64,65]. If an observation is flagged in both DmodX and Hotelling’s T2 as a moderate and strong outlier, it should be excluded. No observation was flagged as a severe outlier, and all observations were thus kept for further downstream statistical analysis.

Beyond the detection of outliers, PCA was used to discover correlation patterns and other similarities in the dataset, including subgroups. The results are displayed in two plots; the score plot and the loading plot. The score plot consists of the first two principal components (PC), and each observation is displayed as an individual point referred to as a score. The loading plot shows each x-variable as a loading weight, and these weights (or loadings) are displayed as individual points in the loading plot. Furthermore, the loading plot shows how each x-variable is related to each other, including if they are positively correlated and contain similar information (located close/grouped), which ones are negatively correlated (located opposite to each other in diagonal quadrants), and variables that are poorly explained by the model (variables located close to zero) [65]. The proteins with the highest loadings have the most influence on the model, and proteins located close to zero do not influence the model output. By combining the score and loading plot, interpretations of associated patterns and correlation structures between variables and observations can be made. Complementary to PCA, the built-in hierarchical clustering analysis (HCA) in SIMCA was used to explore potential subgroups [65]. The resulting dendrogram, based on inputted x-variables (proteins) and all detected components in the PCA, suggested three subgroups. The available clinical variables were explored and compared between subgroups to find potential clinical differences.

#### 2.5.2. Orthogonal Partial Least Square Analysis (OPLS)

OPLS modeling was used to investigate a specific clinical variable (set as the Y-variable) and correlations to x-variables (proteins). OPLS separates the systemic variation in X into two parts, one related or correlated to Y and one uncorrelated to Y. Based on our a priori findings and assumptions, the clinical variables PHQ-9 (depression) and GAD-7 (anxiety) were investigated. Due to an age difference between groups, age was also investigated.

#### 2.5.3. Orthogonal Partial Least Squares Discriminant Analysis (OPLS-DA)

OPLS-DA is a supervised method used to identify variables, such as proteins, that are most important for discriminating between two groups. In this study, OPLS-DA was applied based on the clustered subgroups from the HCA/PCA, where two chronic pain subgroups were set as Y-variables at a time (group belonging) and protein data were set as predictors (x-variables). In the OPLS-DA model, chronic pain patients with moderate/high or mild/no psychological comorbidities were compared. The patients were classified as having moderate or high psychiatric comorbidity if they scored > 10 or mild or no psychological comorbidity if they scored < 10 in the self-reported PHQ-9 and/or GAD-7 scale questionnaires.

The OPLS and OPLSA-DA modeling was performed in two steps, as described in previous studies [32,33]. A number of parameters should be taken into consideration to describe the model quality when using the SIMCA software [64]. Overall model prediction and performance for PCA, OPLS, and OPLS-DA are represented by the variables R^2^ and Q^2^. The *goodness of fit* is how well the model fits the data or *explains* the dataset and is described by R^2^. *The goodness of prediction,* or how well a model predicts the dataset, is explained by Q^2^. These two parameters can vary between 0 and 1, where 1 means a 100% fit and prediction of data. Working with biological data, an R^2^ or Q^2^-value of >0.3 indicates a good model. To evaluate if a model is significant, a cross-validated analysis of variance (CV-ANOVA) is used, where a *p*-value < 0.05 is considered a significant model. To evaluate which x-variables (proteins) are regarded as significant in each model (OPLS and OPLS-DA), the variable influence on projection (VIP)-value is used. VIPpred (VIP value for the first predictive component) is related to the loading of each variable and describes the importance of each variable and its contribution to the model output. The higher the VIP value, the more important the variable for the model is. A VIPpred value > 1.0 is regarded as significant. p(corr) is the correlation coefficient, originating from the loadings and ranging from −1 to 1. A higher p(corr) indicates a higher correlation of a specific variable in OPLS and OPLS-DA models [64,65].

## 3. Results

### 3.1. Cohort Characteristics

In total, 81 subjects with chronic pain were included in this study. Overall, females were overrepresented (71.6%), and the patients reported moderate pain intensity (mean (SD): 6.2 (1.9)) and pain in different areas of the body. The back (70.4%) and neck (58%) were the most reported painful areas (Table 1). Based on the self-reported pain areas, 32.1% were grouped as having local pain, and 67.9% were grouped as having widespread pain (Table 1). An overview of background data, pain, and other clinical characteristics are further described in Table 1.

### 3.2. PCA Investigating Patient Subgroups Based on Inflammatory Protein Profile

In the primary analysis, a PCA was created to investigate correlation structures among inflammatory proteins and potential chronic pain subgroups. In this unsupervised model including 81 observations and 77 proteins, three subgroups of chronic pain patients were identified using HCA included in the SIMCA software. The significant model had four principal components with an R^2^(cum) = 0.420 and a Q^2^(cum) = 0.222 (Figure 2).

The PCA score plot shows each observation (represented by a square, circle, or triangle) and the results from the HCA as three groups marked in the colors green (group 1 = squares), blue (group 2 = circles), and red (group 3 = triangles) (Figure 2A). The complementary PCA loading plot shows each variable (proteins) and is represented by a colored dot (Figure 2B). The different proteins are clustered, correlated, and associated with the three groups. The results from the PCA show a higher inflammatory profile is present in subgroups 2 and 3 compared to subgroup 1 (Figure 2B). Several proteins are intercorrelated and associated with individual subgroups (Appendix A).

To obtain an overview of the clinical picture of the identified three subgroups in the PCA, a descriptive overview of the variables sex, pain intensity, age, BMI, anxiety, and depression scores are presented in Appendix A. In summary, women and men were evenly distributed in the score plot and within each detected subgroup (Appendix A). The self-reported pain intensity was also similar among the three subgroups (Appendix A). A tendency of higher age was noted in subgroup 2 (Appendix A), which was confirmed by statistical analysis showing an increased age in subgroup 2 compared to subgroup 3 (Appendix A). No statistical differences were found between the groups when comparing BMI (Appendix A). A tendency for higher anxiety and depression scores was noted in subgroup 3 (Appendix A). A statistically significant difference in self-reported anxiety score was noted between groups, specifically in subgroup 1 compared to subgroup 3 (Appendix A). For the other clinical measures, we saw no difference in sleep, stress or self-rated health between groups, nor was there a difference in pain-related aspects or physical wellbeing (Appendix A). However, mental wellbeing (MCS-12) differed significantly between groups, with group 2 showing higher overall mental wellbeing as assessed by the SF-12. In summary, the identified groups based on inflammatory markers tended to differ in age, BMI, anxiety, and depression in visual distribution plots (Appendix A) and statistically differed in age, anxiety, and mental wellbeing (Appendix A).

### 3.3. Correlation of Inflammatory Proteins with Pain Intensity, Psychological Comorbidity, and Age

No significant model was obtained when performing an OPLS regression on pain intensity and inflammatory proteins (R^2^ = 0.23, Q^2^ = 0.002, CV-ANOVA *p*-value = 0.91). Since there was a significant difference in self-reported anxiety score (GAD-7) between subgroups 1 and 3, an OPLS model regressing this variable was performed, including all groups. A significant model was obtained with a low explained variation of 14.0% (1 PC, R^2^ = 0.140, Q^2^ = 0.110, CV-ANOVA *p*-value = 0.017) (Appendix A). Ten proteins were associated with anxiety and regarded as significant with a VIP-value > 1.0 and p(corr)> 0.80. These proteins were (ordered after the highest VIP-value): STAMBP, SIRT2, CD40, AXIN1, CASP-8, IL-7, CXCL1, CXCL5, CD244, and ADA (Appendix A). As shown in the score plot (Appendix A), these ten proteins were associated with a higher self-reported anxiety score, where groups 2 and 3 cluster around these proteins, while subgroup 1 does not.

Visualization of the self-reported PHQ-9 scores, reflecting depressive symptoms, in the PCA score plot (Appendix A) indicated a difference in this score among our three subgroups. Therefore, and due to our initial interest in anxiety and depression and comorbidities in pain, an OPLS model using PHQ-9 as a regressor was performed. The significant model had a low explained variation of 14.4% (1 PC, R^2^ = 0.144, Q^2^ = 0.106, CV-ANOVA *p*-value = 0.020), and 11 proteins were regarded as significant and associated with higher self-reported depression scores (VIP-value > 1 and p(corr) > 0.70) (Appendix A). These proteins were (ordered in highest VIP value): STAMBP, SIRT2, AXIN1, CASP-8, ADA, IL-7, CD40, CXCL1, CXCL5, CD244, and ST1A1 (Appendix A).

Ten of the eleven significant proteins were important regressors in both OPLS models of self-reported anxiety and depression scores. Comparing the mean NPX values of these 11 proteins between the 3 subgroups clearly showed that subgroup 1 had lower levels than subgroups 2 and 3 of the 11 cytokines (Figure 3), which was confirmed with univariate statistics (Appendix A).

### 3.4. Inflammation and Age in Chronic Pain

Since there was a significant difference in age between subgroups 2 and 3 (Appendix A), we wanted to control for the possibility that the correlations with psychological comorbidity were not merely an effect of aging. Therefore, an OPLS model using age as an individual y-variable and proteins as regressors (x-variable) was performed to elucidate which proteins correlated to age. For age, a model with high explained variation and predictivity (R^2^ = 0.863, Q^2^ = 0.730, CV-ANOVA *p*-value = <0.001) was found. Twenty-two proteins had a VIP >1.0 and were regarded as significantly correlated to age. Out of the 22 proteins, only the protein ADA was found in the OPLS model of anxiety and depression (Appendix A).

### 3.5. OPLS-DA Modeling of Chronic Pain Patients with a Higher Psychiatric Profile (Anxiety or Depression) Versus Chronic Pain Patients with no Psychiatric Comorbidity

To elucidate if there was a difference in inflammatory profile among chronic pain patients with psychological comorbidities, the cohort was divided into two larger groups, one with moderate/high and one with mild/no psychological comorbidities based on GAD-7 and PHQ-9 scores. The OPLS-DA model was significant (1 PC, CV-ANOVA *p*-value = 0.028) with a fairly good explained variation (R^2^ = 0.279) but with low predictivity (Q^2^ = 0.097) (Figure 4). Several proteins were associated and upregulated in chronic pain patients with the presence of moderate/high psychiatric comorbidity and had high VIP-values (proteins with a VIP > 1.52); IL-7, LAP TGF-beta-1, AXIN1, CXCL1, TNFSF14, CXCL5, CXCL6, 4E-BP1, SIRT2, CD40, CASP-8, ST1A1, STAMBP, ADA, and CD244). Other proteins were more associated with and upregulated in chronic pain patients with mild or no psychiatric comorbidity (proteins with a VIP > 1.0; CDCP1, CXCL9, CXCL10, CCL28, and IFN-gamma). Hence, five of these proteins were also found in the OPLS model of age (ADA, CXCL9, CCL28, CDCP1, and CXCL10). After removing the proteins that were found in the model by age, 15 proteins were left significant and able to separate the two groups (SIRT2, STAMPB, AXIN1, IL-7, CASP-8, CXCL1, CD-40, CXCL5, ST1A1, CD244, TNFSF14, 4E-BP1, CXCL6, LAP TGF-beta-1, and IFN-gamma) (Table 2).

To further elucidate the widespread pain component in this chronic pain cohort, the patients were divided according to their reported pain areas and if they were of a widespread origin or local, if they had back pain or no back pain, and if they had neck pain or neck pain. Surprisingly, no significant OPLS-DA models were retrieved after the division of pain location (data not shown).

## 4. Discussion

In this study, we used an exploratory approach to investigate a set of inflammatory markers in the blood that covary with important aspects of pain syndromes. As a first step, we investigated if the inflammatory markers were related to subgroups in a data-driven approach, i.e., the subgroups were defined according to inflammatory protein aggregation. The data suggest that the studied proteins clustered the participants into three groups. Group 1 had a less pronounced inflammatory profile compared to the other two groups (Figure 2 and Appendix A). The grouping pattern from the PCA is not entirely clear concerning the function of the markers’ biological function. What can be said is that the classical proinflammatory cytokines that are mostly studied in similar research, such as IL-6, IL-8, IFN-gamma, IL-10 and CRP, tend to fall into group 2, except for TNF, which clustered in group 1. To further understand the subgroups identified through the PCA, we compared clinical characteristics between the groups. The identified groups based on inflammatory markers showed a tendency to differ in age, BMI, anxiety, and depression in visual distribution plots (Appendix A), but only age and anxiety distinguished them statistically (Appendix A). Looking at crude mean levels, group 2 distinguished itself by having a higher mean age and BMI and statically significantly better mental health as assessed by SF-12. Group 3 was slightly younger and had the highest anxiety and depression scores. Group 1 had the lowest mean scores for psychological comorbidity and did not stand out in age or BMI. Interestingly, pain intensity and other pain-related measurements did not differ between the identified groups, other than that the older group 2 consequently had suffered from pain for a longer mean time. Sex was evenly distributed in the groups, so we do not believe the identified clustering patterns are sex-dependent. This means that within our PCA, the defining aspects of the groups, other than different inflammatory markers profiles, were age and mental health (Appendix A).

As a next step, we investigated if some markers are specifically related to the three main outcomes of interest, i.e., pain intensity, anxiety, and depression. First, we concluded that no inflammatory markers were related to pain intensity. This was somewhat contrary to our expectations, given prior research on inflammatory markers in pain populations. However, previous research is incongruent regarding the correlation of protein markers with pain intensity [34,35,36,37,38,66,67]. The lack of correlation to pain intensity was perhaps also not surprising, given that the PCA did not show any differences in pain measures in the identified groups. In contrast, 11 proteins were associated with higher depression scores. The same proteins, except for ST1A1, were also significantly correlated to higher anxiety scores. Group 1 had significantly lower mean levels of these markers compared to the other two groups (Figure 3). Unfortunately, we could not find any articles studying pain and depression or anxiety using this inflammatory panel for comparison. However, Sundquist et al. [68] showed altered levels of 36 cytokines/chemokines in patients with mild to moderate depression and anxiety after 8 weeks of psychological treatment using the same inflammatory panel. Among their top 10 most significant proteins that were altered after treatment were the same proteins that we found in our OPLS and OPLS-DA models for depression and anxiety (CXCL6, STAMPB, CXCL1, SIRT2, AXIN1, CXCL5, ST1A1, and IL-7). Even though they did not find any correlation of the cytokines/chemokines to reported symptom severity (using MADRS-S) before the start of the intervention, these proteins decreased after the intervention, and the authors suggest a potential involvement of these inflammatory markers in depression, anxiety, and stress and adjustment disorders. Our subgroup analysis partly supports that there might be specific inflammatory protein networks potentially related to self-reported higher anxiety and/or depressive mood in chronic pain patients.

To investigate if the correlations were primarily driven by age, given the significant age difference of one group, we identified the proteins that correlated with age (Appendix A). From these 22 age-related markers, only 1 (ADA) overlapped with the markers related to OPLS models of anxiety and depression. We thus tentatively argue that the correlations suggest a relationship between some immunomarkers and psychological comorbidity in this pain cohort. Taking this reasoning a step further, we divided the patients into two groups, pain with high psychological comorbidity vs. pain with low psychological comorbidity, and tested if some markers were predictive for this division. When excluding the markers that were related to age (CXCL9, CXCL10, CDCP1, CCL28, and ADA), 15 markers had some prognostic capacity (Figure 4 and Table 2), of which the 11 from the OPLS were overlapping. Interestingly, several of the markers associated with the patients in our cohort who also suffer from comorbid psychological problems have been associated with the presence of comorbid pain in patients groups with a different primary diagnosis, including diabetes [24] and myalgic encephalomyelitis/chronic fatigue syndrome (ME/CFS) [69], and have been shown to add inflammation in the form of periodontal problems in rheumatoid arthritis [70]. Several of these markers have also been suggested as markers for a specific pain disorder [71,72,73], such as AXIN1, CXCL1, TNFSF14, CD40, and particularly ST1A1 (all three studies). Additionally, Fineschi et al. [74] showed in a recent study of fibromyalgia patients that among 19 increased inflammatory proteins, AXIN1, SIRT2, and STAMPB were among the highest elevated in FM compared to controls. They also identified a subgroup of FM patients presenting a “high inflammatory profile” based on inflammatory protein levels in their serum. This FM subset reported significantly higher FM severity scores compared to the FM subset with a less pronounced inflammatory protein profile. However, no correlation between inflammatory proteins and depression or anxiety scores was assessed [74]. Some of these markers are reoccurring in several studies, and it remains to be understood what their presence in the blood in complex disorders signifies. AXIN1, for example, a regulator molecule of the Wnt signaling pathways [75], has been shown to correlate with disease severity in endometriosis [76]. SIRT2 has a wide range of regulatory functions ranging from neuronal to immunological and metabolic homeostasis [77,78]. Our study, together with other protein disease signature (PDS) studies, shows the importance of analyzing multiple inflammatory markers and comorbidities to explore a more complete picture of a disease. Our type of analysis strategy has the potential to reveal markers that would not have been studied in a classical hypothesis-driven setting.

Interestingly, all 15 markers that, in our OPLS-DA model, were related to the presence of anxiety or depression were also part of a highly discriminative PDS for fibromyalgia proposed by Bäckryd et al. [79]. In our study, post hoc analyses did not show a difference between patients with localized versus widespread pain. Speculatively, in a complex disorder such as chronic pain, any disease-related PDS that is identified may not be specific for the disease characteristics of primary interest but with characteristic comorbidity. This would require a new approach when comparing groups and controlling for covariates. The markers in the mentioned studies above were identified in between-group analyses of a particular pain population and healthy control or between two pain populations. In such comparisons, it seems to be assumed that the identified markers are distinct for the pain syndrome at hand, while psychological comorbidity is rarely taken into the equation. Our results suggest that these markers may, in fact, be related to pain syndromes but perhaps not to pain-related aspects specifically or to specific pain syndromes. PDS exploration needs to take psychological comorbidity into account to a higher degree, e.g., with positive controls with psychiatric disorders or grouping based on mental health or other common comorbidities compared to disease-specific severity.

Our study has several limitations. The PDS was assessed in a mixed chronic pain population representative of patients attending this clinic working with behavioral interventions for chronic pain. The group appears typical for similar clinical settings [33,80], with a mean age of around 50 years, around 70% women, and common comorbidities (Table 1), with the majority of patients having low back and neck pain and pain in more than one body part. This may give the results some ecological validity but introduces variance in the sample, and the results must be interpreted keeping this in mind. We lack a healthy control group for comparison, and a positive control group, such as a group of depressed patients. The design is cross-sectional, and the variables are self-reported. The chosen analysis panel is inflammatory, which constricts the exploratory effort, and the values are relative. We have not included several variables that may be of importance for the relationships at hand, such as medication, socioeconomic situation, and comorbidities other than anxiety and depression. Furthermore, the blood is not the optimal target tissue from a mechanistic point of view, but it is the most accessible sample for clinicians. Cerebrospinal fluid (CSF) may be more relevant mechanistically [81], and there is not a clear correlation between peripheral and central cytokine expression [82]. However, CSF is more challenging to retrieve, and collection may pose a risk for the participants that need to be weighed against the benefits and scientific merit. From a psychoneuroimmunological point of view, however, one may argue that peripheral activity may indeed affect the central nervous system and, consequently, emotions and behaviors [83]. Finally, correlations are not informative for causation, but such relationships can give an indication of which systems may covary and what active networks characterize the studied patient group. Despite these limitations, there are some clear overlaps between our findings and those of prior studies using the same panel in pain populations or populations with comorbid pain, which may suggest that there are overarching biological networks that need to be identified to be able to specify disease-related networks.

In our sample, the Olink panel only gives relative relationships, but the CRP levels suggest that none of our included patients had severe inflammation, and most of them had normal levels. The term that is often used for non-communicable complex disorders is “low-grade inflammation”, but this term needs to be further problematized. Within a population defined as having low-grade inflammation, relationships and characteristics in subgroups of patients with higher or lower inflammation may differ. Furthermore, ongoing inflammatory activity may not be harmful but a part of the body’s attempt to heal. Recently, we showed that in pelvic pain, women had an inverse relationship between the widespreadness of the pain and blood IL-8 levels, whereas no association was seen for men [84]. Interestingly, a very similar pattern was shown for depressed patients [85], with an inverse relationship between blood IL-8 levels and depression severity scores, but no association was found for men. Furthermore, the key to immune-driven ill health may not be systemic or localized inflammation per se but dysfunctional immunoregulation. Regarding pain specifically, some recent data show a stronger association between pain development and immunoreactivity [86,87], as gauged by toll-like receptor stimulation of whole blood/white blood cells, than with ongoing inflammatory activity (cytokine plasma levels). We did not have the logistic capabilities to collect and stimulate white blood cells in this study, but it is possible that immune priming and second-hit regulation [1] may be of greater importance for persistent pain than the commonly measured inflammatory markers assessed in blood.

Our results should be seen as a complement to ongoing neuroimmune research in pain and potentially hypothesis-generating rather than mechanistically exploratory. We argue that the pain and PNI research communities need to move beyond the traditional inflammatory markers to understand complex disorders, consonant with beliefs about multifaceted and interacting bodily processes involved in health and disease. IL-6 and CRP have repeatedly been shown to correlate with self-rated health [88,89], psychological wellbeing [19,21], and pain sensitivity [39,90]. These findings have taught us that low-grade inflammation is important, but from a treatment perspective, such general markers are not particularly useful. Treatment strategies targeting, e.g., IL-6 in these populations would do more harm than good. While anti-cytokine treatment has been successful for many patients with inflammatory pain, such as in rheumatoid arthritis [14], similar successes for musculoskeletal and generalized pain have not been achieved. It is likely that other proteins that interact and regulate the (neuro)immune system will be better candidates to treat pain with no clear inflammatory origin and may influence the large variability in treatment outcomes.

## 5. Conclusions

This study further contributes to the increasing knowledge that inflammation seems to be involved in the underlying mechanisms of chronic pain conditions. Our results show that there are potential subgroups of chronic pain patients with different inflammatory profiles and that specific proteins might be associated with higher psychological comorbidity. In future research, inflammatory blood signature exploration needs to consider psychological comorbidity to a higher degree to further understand its involvement in different chronic pain conditions.

## Figures and Tables

**Figure 1 biomedicines-11-00713-f001:**
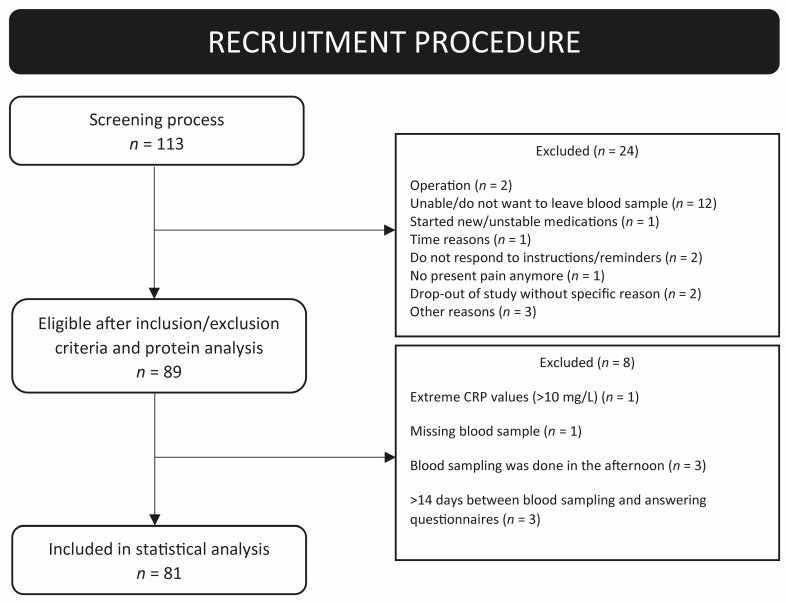
Scheme of recruitment.

**Figure 2 biomedicines-11-00713-f002:**
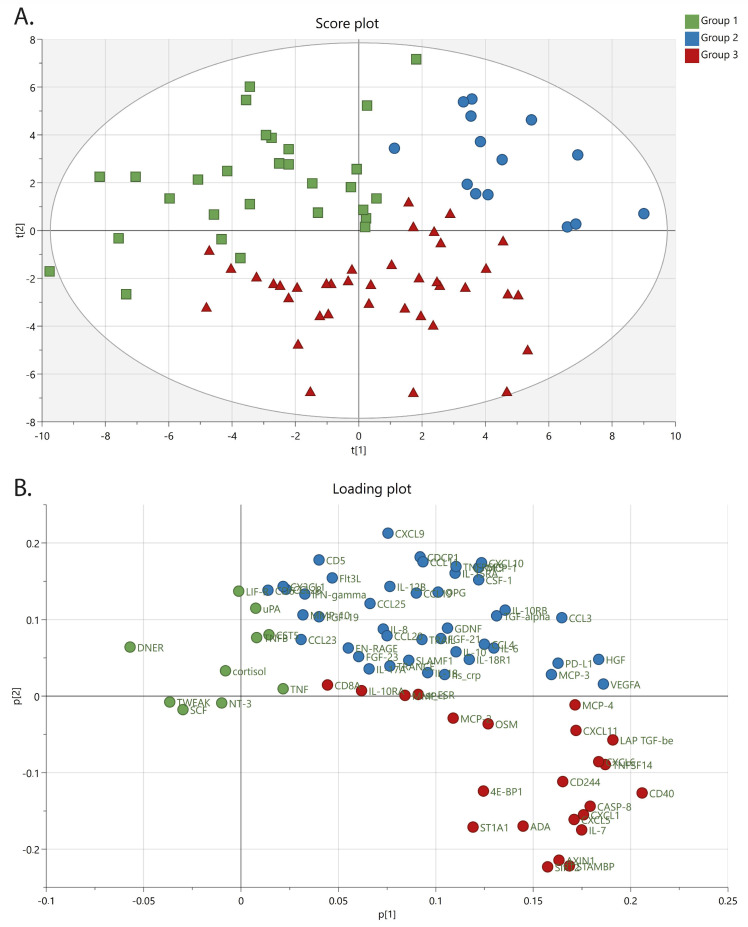
PCA of inflammatory proteins, including subgroups after HCA. (**A**) Score plot showing observations (each individual subject). Colors are according to the three groups identified by HCA. Squares = group 1 (green), circles = group 2 (blue), triangles = group 3 (red). t (1) and t (2) represent the first and second principal components, respectively. (**B**) Loading plot showing proteins (x-variables) and associated clusters of proteins for each group. Colors on dots correspond to colors in score plot and respective group. p (1) and p (2) represent the first and second principal components, respectively.

**Figure 3 biomedicines-11-00713-f003:**
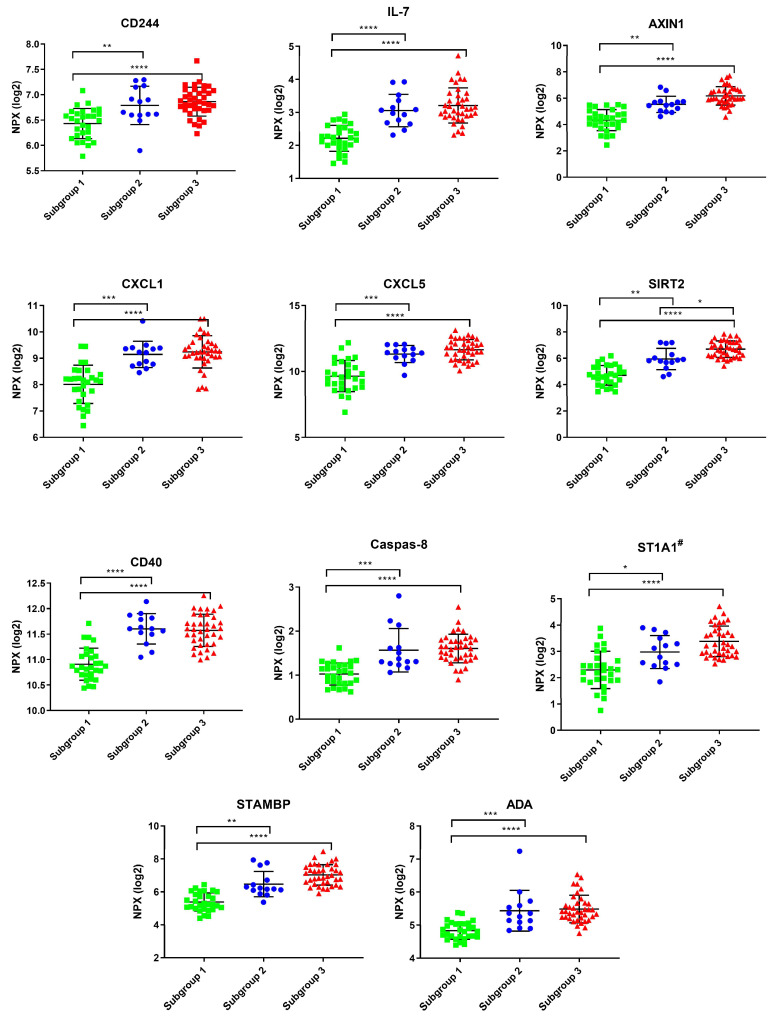
Mean Log2 NPX values of significant proteins correlated with PHQ-9 and GAD-7. Groups are based on HCA clustering from the PCA model only including proteins. # Only significant in the OPLS model of PHQ-9. * *p*-value < 0.05, ** *p*-value < 0.01, *** *p*-value < 0.001, **** *p*-value < 0.0001. CD244 = natural killer cell receptor 2B4; IL-7 = interleukin-7, AXIN1 = axin-1; CXCL1 = growth-regulated alpha protein; CXCL5 = C-X-C motif chemokine 5; SIRT2 = NAD-dependent protein deacetylase sirtuin-2; CD40 = tumor necrosis factor receptor superfamily member 5; CASP-8 = caspase-8; ST1A1 = sulfotransferase 1A1; STAMBP = STAM-binding protein; ADA = adenosine deaminase.

**Figure 4 biomedicines-11-00713-f004:**
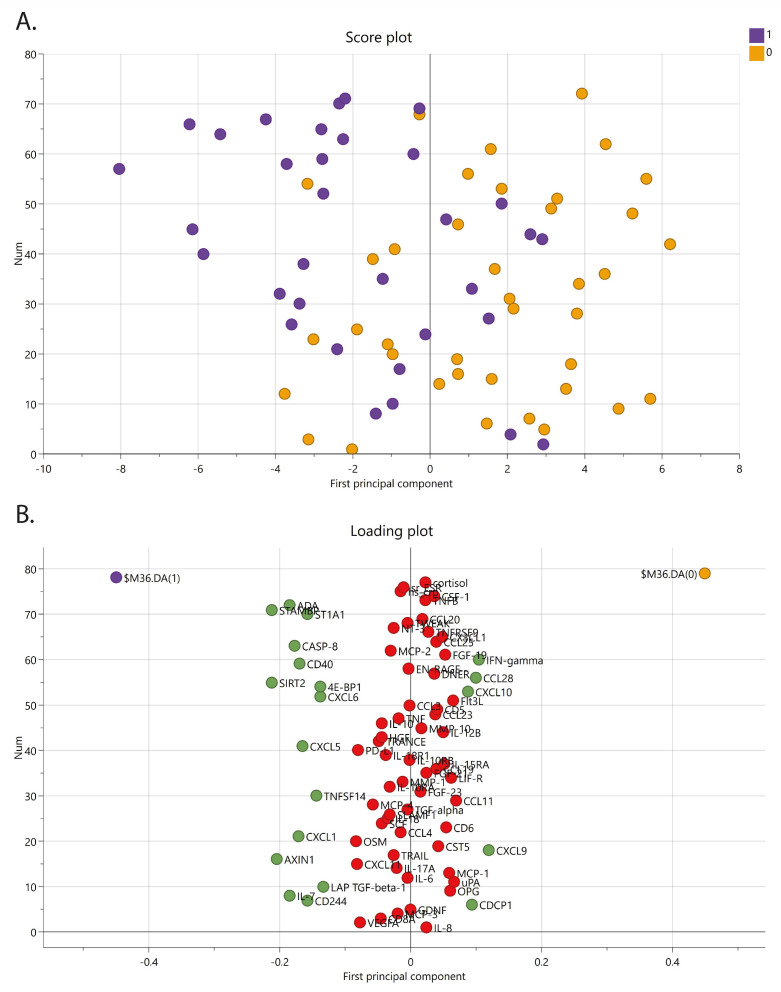
OPLS-DA of chronic pain patients with psychological comorbidities of anxiety and depression. (**A**) Score plot, 1 (purple) = chronic pain patients with moderate/high psychological comorbidity scoring >10 on PHQ-9 and/or GAD-7 scale, and 0 (orange) = chronic pain patient with mild or no presence of anxiety or depression (scoring <10 on PHQ-9 and/or GAD-7 scale). (**B**) Loading plot showing significant proteins (green dots are proteins with a VIP > 1.0 and regarded as significant proteins. Red dots are non-significant proteins (VIP < 1.0).

**Table 1 biomedicines-11-00713-t001:** Background and clinical data. These data have been published elsewhere, although not with the same number of participants [52].

Variables	Full Cohort (*n* = 81)
Background data	Mean (SD)
Age (years)	51.0 (14.6)
BMI (kg/m ^2^ )	25.3 (4.5)
Sex females/males % (n)	71.6% (58)/28.4% (23)
Days difference blood sample and filling in questionnaires	2.6 (2.8)
Pain characteristics	
Pain area number (sum)	4.7 (2.9)
Local pain ( 1–2 sites)	32.1%
Widespread pain (>3 sites)	67.9%
Type of pain (reported pain in following areas)	
Back	70.4%
Neck	58.0%
Leg	51.9%
Headache	48.1%
Foot	45.7%
Arm	40.7%
Hand	35.8%
Abdominal	34.6%
Entire body	32.1%
Teeth and jaw	23.5%
Chest	11.1%
Facial	9.9%
Genital	6.2%
Pain intensity (NRS)	6.2 (1.9)
Pain duration (years)	11.3 (8.9)
Psychological comorbidity	
Depression (PHQ-9)	9.3 (5.4)
Anxiety (GAD-7)	6.3 (4.5)
Other clinical parameters	
Insomnia (ISI)	13.3 (6.6)
Stress (PSS)	20.0 (9.1)
General health (SRH-5)	2.9 (1.0)
Physical wellbeing (PCS-12)	33.0 (8.6)
Mental wellbeing (MCS-12)	39.8 (13.2)

**Table 2 biomedicines-11-00713-t002:** Significant proteins of OPLS-DA model of psychiatric profile in chronic pain patients. Proteins with a VIP value > 1.0 are considered significant, and the mean (standard deviation) is presented for each group. Proteins correlated to age are not shown. Moderate/high psychiatric comorbidity score >10 on PHQ-7/GAD-9, and mild/no psychiatric comorbidity score <10 on PHQ-7/GAD-9.

Protein Name	Uniprot Protein Accession Number	VIP	p(corr)	Moderate/High Psychological Comorbidity Mean (SD) (*n* = 33)	Mild/No Psychological Comorbidity Mean (SD) (*n* = 39)
**SIRT2**	Q8IXJ6	2.42	−0.90	6.28 (0.98)	5.58 (1.05)
**STAMBP**	O95630	2.42	−0.90	6.71 (0.88)	6.12 (0.92)
**AXIN1**	O15169	2.32	−0.86	5.84 (1.01)	5.23 (1.02)
**IL-7**	P13232	2.09	−0.78	3.09 (0.63)	2.70 (0.58)
**CASP-8**	Q14790	2.02	−0.75	1.51 (0.47)	1.32 (0.40)
**CXCL1**	P09341	1.94	−0.72	8.99 (0.73)	8.71 (0.92)
**CD40**	P25942	1.94	−0.72	11.48 (0.41)	11.30 (0.46)
**CXCL5**	P42830	1.88	−0.70	11.30 (1.02)	10.74 (1.25)
**ST1A1**	P50225	1.79	−0.67	3.17 (0.82)	2.84 (0.76)
**CD244**	Q9BZW8	1.79	−0.66	6.81 (0.34)	6.66 (0.36)
**TNFSF14**	O43557	1.64	−0.61	3.98 (0.55)	3.65 (0.48)
**4E-BP1**	Q13541	1.57	−0.58	9.45 (0.82)	9.18 (0.93)
**CXCL6**	P80162	1.56	−0.58	8.78 (0.64)	8.41 (0.98)
**LAP TGF-beta-1**	P01137	1.52	−0.57	7.30 (0.37)	7.07 (0.40)
**IFN-gamma**	P01579	1.19	0.44	5.74 (0.62)	6.50 (1.19)

## Data Availability

The data will not be available for open access due to sensitive information and ethical restrictions, but please contact the corresponding author for access.

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
