# Peer review of "Inflammatory Blood Signature Related to Common Psychological Comorbidity in Chronic Pain"

_biomedicines, 2023, doi:10.3390/biomedicines11030713_

Round 1
Reviewer 1 Report
The authors conducted a study to investigate whether inflammatory blood signatures are associated with intensity of pain or psychological comorbidity in a sample including 81 patients with chronic pain.
The investigated topic is of interest as chronic pain and its comorbidities are a relevant source of disability. The manuscript is well written and easy to read. The rationale of the study is well explained and the introduction is adequately referenced. The methods are sound and the conclusions are supported by results. I have particularly appreciated the detailed information regarding inclusion criteria and screening procedures. Findings of the study are well presented, also using tables and figures, and discussed in the context of previous literature. I think this paper adds to the literature on the topic and is worth of publication. I only have minor comments.
- It was not clear to me whether correction for multiple testing comparison was applied. This should be better specified in the methods and, in case no correction was applied, this decision should be motivated
- Table 2 should be prepared as a table rather then a screenshot of an excel file (the screenshot is also reporting marks related to Office grammar revision tools that should be removed)
Author Response
Thank you for giving us the opportunity to revise our manuscript. We thank the reviewers for the encouraging feedback. All suggestions have been implemented, and changes are visible via the tracking function in Word.
Reviewer 1:
Regarding multiple comparisons, this has been adjusted for when investigating the patient reported outcomes and specific markers between the three identified subgroups (supplementary table 2 and 3), using Kruskal-Wallis test and the post hoc Dunn's multiple comparisons test. This is explained the following sentence line 273-276 …”For comparison between subgroups (PROMs and specific protein markers), the Kruskal-Wallis test followed by post hoc Dunn’s multiple comparison test were applied. No further correction for multiple testing was performed for univariate statistics”.
Multivariate analyses are designed to handle these issues as we are analyzing so many variables at once in omics approaches. We have added a sentence about this in the manuscript, to help the readership of Biomedicines to evaluate the findings sufficiently. See line 280-285 …” MVDA is often used when large omics panels are investigated, and multivariate correlation between variables exists. Since MVDA accounts for the internal correlation structures of a data set (analyzing all variables simultaneously), hence, no further correction for multiple testing is needed, as is often necessary using other type of tests such as mixed models or univariate comparison.”….
Reviewer 2:
We have shortened the abstract and amended English grammar mistakes. We have added a relevant reference as suggested (reference 20). Some of the problems with the figures and tables (Reviewer 2) were due to us having some issues with the MDPI format template, we apologize for this. We have changed this in the revised manuscript, and we have also uploaded all figures, tables and figure texts as separate documents. The reference list should be according to the Endnote MDPI format now.
Sincerely,
Bianka Karshikoff
Karin Wåhlén
Jenny Åström
Mats Lekander
Linda Holmström
Rikard K. Wicksell
Reviewer 2 Report
First of all, I would like to congratulate the authors on a quality piece of research.
The abstract is too long, please reduce contents according to the author's guidelines and try to save the message.
- Line 18 due to “a” lack of
- Line 27 subgroups 2 and 3
- Such little grammar errors occur throughout the article, please revise
I have seen some recent articles in the International Journal of Molecular Sciences reviewing cytokines and neuropeptides in inflammation, depression, and anxiety also linked to chronic pain through the HPA axis. Maybe include these citations in the article to help the fellow journal.
- Please revise all figures to a better resolution (either save as a tiff format in PowerPoint or a different “lossless” format with at least 300dpi in the software you use) because they get very pixelated and hard to read, especially figures 2 and 4
- Also, table 2 needs to be a table, not a figure, please revise
References
- The citation style is not according to MDPI, please revise using endnote, Zotero, or a similar bibliography tool (only a cosmetic issue)
Author Response
Dear Editor and Reviewers,
Thank you for giving us the opportunity to revise our manuscript. We thank the reviewers for the encouraging feedback. All suggestions have been implemented, and changes are visible via the tracking function in Word.
Reviewer 1:
Regarding multiple comparisons, this has been adjusted for when investigating the patient reported outcomes and specific markers between the three identified subgroups (supplementary table 2 and 3), using Kruskal-Wallis test and the post hoc Dunn's multiple comparisons test. This is explained the following sentence line 273-276 …”For comparison between subgroups (PROMs and specific protein markers), the Kruskal-Wallis test followed by post hoc Dunn’s multiple comparison test were applied. No further correction for multiple testing was performed for univariate statistics”.
Multivariate analyses are designed to handle these issues as we are analyzing so many variables at once in omics approaches. We have added a sentence about this in the manuscript, to help the readership of Biomedicines to evaluate the findings sufficiently. See line 280-285 …” MVDA is often used when large omics panels are investigated, and multivariate correlation between variables exists. Since MVDA accounts for the internal correlation structures of a data set (analyzing all variables simultaneously), hence, no further correction for multiple testing is needed, as is often necessary using other type of tests such as mixed models or univariate comparison.”….
Reviewer 2:
We have shortened the abstract and amended English grammar mistakes. We have added a relevant reference as suggested (reference 20). Some of the problems with the figures and tables (Reviewer 2) were due to us having some issues with the MDPI format template, we apologize for this. We have changed this in the revised manuscript, and we have also uploaded all figures, tables and figure texts as separate documents. The reference list should be according to the Endnote MDPI format now.
Sincerely,
Bianka Karshikoff
Karin Wåhlén
Jenny Åström
Mats Lekander
Linda Holmström
Rikard K. Wicksell
Round 2
Reviewer 2 Report
- The article has changed sufficiently to be published.